# Energy Harvesting Maximizing for Millimeter-Wave Massive MIMO-NOMA

**Shufeng Li *[ID], Zelin Wan, Libiao Jin and Jianhe Du**

School of Information and Communication Engineering, Communication University of China, Beijing 100024, China; 13787218609@163.com (Z.W.); libiao@cuc.edu.cn (L.J.); dujianhe1@163.com (J.D.)
* Correspondence: lishufeng@cuc.edu.cn; Tel.: +86-1355-2288-205

**Abstract:** Multiple-Input Multiple-Output Non-Orthogonal Multiple Access (MIMO-NOMA) is considered a promising multiple access technology in fifth generation (5G) networks, which can improve system capacity and spectral efficiency. In this paper, we proposed two methods of user grouping and proposed a dynamic power allocation solution for MIMO-NOMA system. Then we proposed an algorithm to maximize energy harvest for MIMO-NOMA system by integrating Simultaneous Wireless Information and Power Transfer (SWIPT), known as maximizing energy harvesting. Specifically, we added a power splitter at the receiver and found the optimal power splitting factor for each user. The harvested power of the user is maximized under the premise of satisfying the minimum communication rate. The simulation results show that the proposed method is effective.

**Keywords:** MIMO; NOMA; precoding; power allocation; user-clustering; power splitter

## 1. Introduction

Non-Orthogonal Multiple Access (NOMA) is one of the key technologies of fifth generation (5G) networks, which can significantly improve the overall performance of the system. Through time and frequency resource reuse and user grouping for large-scale connection, NOMA can improve spectral efficiency [1]. Multiple-Input Multiple-Output (MIMO) is regarded as a promising technique for 5G wireless communication systems. The principle of MIMO is the use of multi-antenna technology to achieve spatial diversity. The multi-receiving and multi-transmitting mechanisms can effectively combat multipath interference and increase system capacity. The author proposed a new low complexity arrival direction estimation algorithm in MIMO system for meeting the needs of green communication in [2]; this algorithm is based on a new downlink transmission frame structure that can make full use of the prior information under the channel codebook feedback mechanism. NOMA superimposes multiple user signals in the power domain, superimposes coding on the transmitter, performs serial interference cancellation on the receiver, and eliminates inter-user interference for grouped users [3]. NOMA can be divided into two categories: NOMA in power domain and NOMA in code domain. The power domain NOMA scheme can provide service to multiple users with different channel conditions simultaneously in the same time, frequency, coding, and space [4–7]. Uplink and downlink NOMA transmission of single cellular network is studied in [8], the author also analyzed the effect of distance on performance of the system. The influencing factors of NOMA, such as user power allocation, new order cost, and Serial Interference Cancellation (SIC) error propagation were discussed in [9].

NOMA combined with MIMO technology has attracted considerable research interest. The basic principle of NOMA and MIMO combination in downlink transmission was studied in [10]. The MIMO-NOMA could improve spectrum reuse efficiency, transmission throughput, and energy efficiency. In MIMO-NOMA, it is essential to make user grouping efficient. If the user is grouped by

appropriate methods, the error rate of the system can be reduced [11]. In the existing MIMO-NOMA system, users are divided into multiple groups. The group uses the NOMA principle to serve users. The precoding between groups is used to eliminate interference. The user grouping of the downlink NOMA system is studied in [12]; the user clustering problem is formalized into a semi-definite programming problem which can be solved using numerical toolbox [13]. The accuracy of power allocation affects the system performance. In [14], the author derived the closed expressions for the traversal, rate, and interrupt probability of the two-user NOMA system for static power allocation. A dynamic power allocation solution is provided in [15] with the goal of maximizing the total unit capacity.

Wireless communication devices still use electric cables or batteries to obtain electric energy. The battery storage capacity and usage period are often limited, which will cause the development and application of new technologies in specific scenarios to be deeply bounded. Harvesting energy from radio frequency (RF) signals has become an attractive strategy to address the critical challenges of limited battery life in wireless communication networks. An advanced technology called Simultaneous Wireless Information and Power Transfer (SWIPT) emerged in [16], by which the energy transmission and information transmission using RF signals can be achieved. Therefore, SWIPT is considered a potential energy-saving solution for 5G [17], which has attracted widespread attention in academia and industry. A capacity-energy function was defined and the receiver can perform both information decoding and energy harvesting (EH) without any restrictions [16]. The work [18] considered the sum rate and the per-user optimized data rate of the SWIPT-enabled NOMA system, in which two information decoding schemes are proposed, "fixed decoding order" and "time sharing", respectively, and proved that system performance could be significantly improved by integrating SWIPT on NOMA. The work of [19] jointly optimized the transmission power of the Base Station (BS), as well as the length of time for energy acquisition and data transmission. The application of SWIPT technology in NOMA is studied and a new cooperative SWIPT NOMA protocol is proposed in [20]. By jointly optimizing the power allocation coefficient of MIMO-NOMA and the power splitting factor of SWIPT, the achievable sum rate can be maximized [21].

The issue of energy conservation is also an issue that recently has attracted considerable attention. To meet the needs of green communication and realize the recycling of energy, we implement energy-saving wireless communication in MIMO-NOMA system integrated with SWIPT. Specifically, each user uses a power splitter to split the received signal into two parts. The receiver performs information retrieval and energy storage to implement SWIPT simultaneously. In this paper, we also studied the user clustering, precoding design, and power allocation to optimize the power-splitter factor of SWIPT. The harvested energy is maximized under the premise of satisfying the minimum communication rate of the user.

The main contributions of this paper are as follows:

1.  In this paper, for reducing intra-cluster interference and complexity, we proposed two methods for user grouping. One is based on channel gain; the other is based on antenna grouping. The performance effects of the two grouping methods on the system are analyzed from the perspective of spectrum efficiency and energy efficiency.
2.  We provided a dynamic power allocation solution for downlink multi-user MIMO-NOMA. Power allocation is divided into two steps: power allocation between clusters and power allocation within the cluster. In the power allocation between clusters, the power allocation of each beam is proportional to the number of users, and the power within the cluster is allocated according to the maximum communication rate of the cluster.
3.  We proposed the energy harvesting maximizing method. We added a SWIPT split receiver in MIMO-NOMA system for each user. In this method, the optimal power split coefficient of each user is found with the optimization objective of harvesting the maximum energy and satisfying the minimum communication rate of users.

*Notation*: Within this paper, the upper-case boldface letters denote matrices; lower-case boldface letters are vectors. $(\cdot)^T, (\cdot)^{-1}, (\cdot)^H$ denotes the transpose, matrix inversion, and conjugate transpose. $CN(a, b)$ denote the complex Gaussian distribution with mean a and covariance b. $\|\cdot\|_n$ is $l_n$ norm operation. $|\Gamma|$ denotes the number of elements in set $\Gamma$. $E\{\cdot\}$ denotes the expectation.

## 2. System Model

In this paper, we consider a single-cell downlink Millimeter-Wave massive MIMO-NOMA system, as shown in Figure 1. The base station is equipped with $N$ antennas and $N_{RF}$ RF chains, and $K$ single antenna users are served by the base station. By using NOMA, each beam can support multiple users. $S_g$ represents the set of users served by the $g$th beam. To fully realize the multiplexing gain, we assume that the beam number $G$ is equal to the number of RF chains $N_{RF}$. The received signal at $m$th user in the $n$th beam is [21]:

$$
\begin{aligned}
y_{n,m} &= \mathbf{h}_{n,m}^H \sum_{j=1}^{G} \sum_{i=1}^{|S_n|} \mathbf{w}_j \sqrt{p_{i,j}} s_{i,j} + v_{n,m} \\
&= \mathbf{h}_{n,m}^H \mathbf{w}_n \sqrt{p_{n,m}} s_{n,m} + \mathbf{h}_{n,m}^H \mathbf{w}_n \sum_{j=1}^{m-1} \sqrt{p_{n,j}} s_{n,j} + \mathbf{h}_{n,m}^H \mathbf{w}_n \sum_{i=m+1}^{|S_i|} \sqrt{p_{n,i}} s_{n,i} \\
&\quad + \mathbf{h}_{n,m}^H \sum_{i \neq n}^{G} \sum_{j=1}^{|S_i|} \mathbf{w}_j \sqrt{p_{i,j}} s_{i,j} + v_{n,m}
\end{aligned}
\tag{1}
$$

where the interference within the cluster and the interference between the cluster are existing. $\mathbf{h}_{m,n}$ represents the channel of the $m$th user in the $n$th beam, $\mathbf{w}_n$ is precoding vector of the $n$th beam. $s_{n,m}$ is the transmitted signal and $p_{n,m}$ denotes transmitted power for the $m$th user in the $n$th beam, and $v_{n,m}$ is the noise following the distribution $CN(0, \sigma_u{}^2)$.

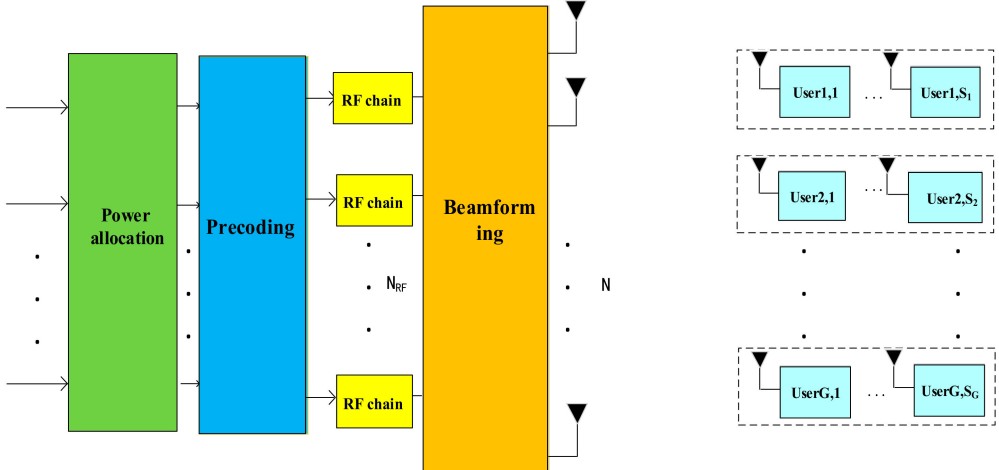

**Figure 1.** Millimeter-Wave massive Multiple-Input, Multiple-Output, Non-Orthogonal Multiple Access (MIMO-NOMA) system.

The $m$th user in the $n$th beam can eliminate the interference of the $i$th user (for all $i > m$) in the $n$th beam by performing Serial Interference Cancellation (SIC). The remaining signal received by the $m$th user in the $n$th beam can be rewritten as:

$$
\widetilde{y}_{n,m} = \left( \mathbf{h}_{n,m}^H \mathbf{w}_n \sqrt{p_{n,m}} s_{n,m} + \mathbf{h}_{n,m}^H \mathbf{w}_n \sum_{j=1}^{m-1} \sqrt{p_{n,j}} s_{n,j} + \mathbf{h}_{n,m}^H \sum_{i \neq n}^{G} \sum_{j=1}^{|S_i|} \mathbf{w}_j \sqrt{p_{i,j}} s_{i,j} + v_{n,m} \right)
\tag{2}
$$

The Signal to Interference plus Noise Ratio (SINR) at the *m*th user in the *n*th beam is:

$$\gamma_{n,m} = \frac{\left\|\mathbf{h}_{n,m}^H \mathbf{w}_n\right\|_2^2 p_{n,m}}{\xi_{n,m}} \tag{3}$$

where,

$$\xi_{n,m} = \left\|\mathbf{h}_{n,m}^H, \mathbf{W}_n\right\|_2^2 \sum_{j=1}^{m-1} p_{n,j} + \sum_{i \neq n}^{G} \left\|\mathbf{h}_{n,m}^H \mathbf{W}_i\right\|_2^2 \sum_{j=1}^{|S_i|} p_{i,j} + \sigma_v^2 \tag{4}$$

The achievable rate of the *m*th user in the *n*th beam can be written as:

$$R_{n,m} = \log_2(1 + \gamma_{n,m}) \tag{5}$$

Finally, the achievable sum rate is:

$$R_{sum} = \sum_{n=1}^{G} \sum_{m=1}^{|S_i|} R_{m,n} \tag{6}$$

## 2.1. User-Clustering

We assume that all users in the downlink MIMO cellular system can utilize NOMA-based resource allocation. Users need to be grouped first and the grouped users share a set of codes in the same group by the precoding matrix. Low channel gain users of NOMA clusters are often subject to higher intra-cluster interference [22].

In this paper, we propose the following two methods for user clustering:

(1) The user clustering method based on channel gain. As mentioned in [23], the cluster head user with the highest channel gain can eliminate intra-cluster interference, thereby obtaining the maximum throughput gain. Therefore, the keys to maximize overall system capacity is to ensure that high channel gain users are selected as cluster heads for different MIMO-NOMA clusters in one unit. To improve system performance, we grouped users by assigning the user with largest channel gain as cluster head. As shown in Figure 2, the number of user groups *M* is equal to the number of beams *G*. In this way, users in the same group will suffer higher channel correlation, which is beneficial to eliminate interference between users. The lower equivalent channel correlation of users in different beams is beneficial to eliminate inter-beam interference, which improves the multiplexing gain. The proposed solution is described in Algorithm 1.

---

**Algorithm 1**: User Clustering based on Channel Gain (UC_CG)

---

Input:
The number of users *K* and the number of antennas *N*;
Channel vectors: $\mathbf{h}_k$ for $k = 1, 2, \cdots K$, $N_{RF}$
Output:
User-grounding T
1.Select number of cluster-heads;
$\mathbf{H} = [|\mathbf{h}_1|, |\mathbf{h}_2|, \cdots, |\mathbf{h}_K|]$, $[\sim, order] = (sort(H), 'descend')$
$O = [order\,(1), \cdots, \,order\,(G)]$
2. Include other users into each cluster;
$O^C = K/O$;
$\max\left|\mathbf{h}_i^H \mathbf{h}_j\right|, \forall i \in O, j \in O^C$, Grouping the cluster channel with a large correlation.
Return T

---

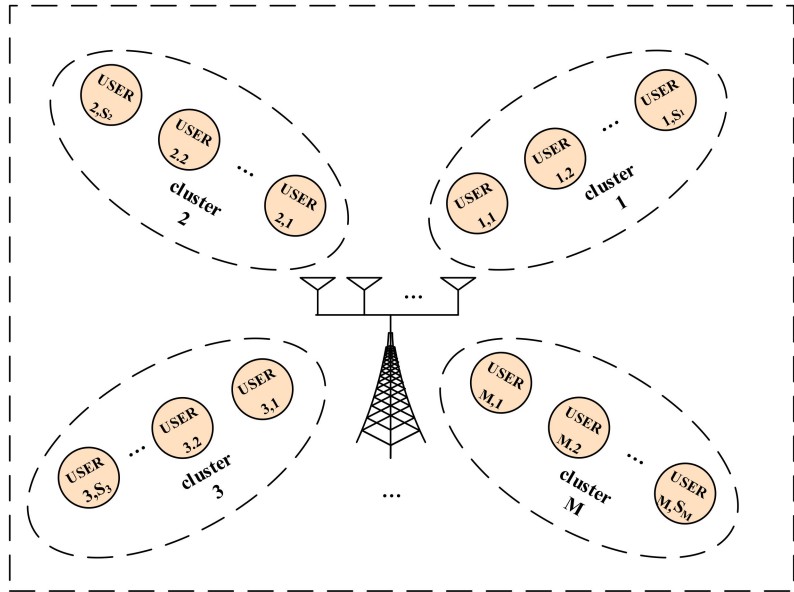

**Figure 2.** User clustering based on channel gain.

(2) The user clustering method based on antenna grouping. We consider the downlink MIMO-NOMA system, the number of users $K$ is larger than the number of beams $G$. We provide a low complexity MIMO-NOMA user clustering algorithm, where the number of clusters $G$ is equal to the number of RF chains $N_{RF}$. As shown in Figure 3, the antennas at the BS are sequentially grouped into $G$ groups, there are $Nt$ antennas in each group. We first select the user with the largest channel gain corresponding to each antenna group as the cluster head and find the correlation between the remaining users and each cluster head user. Then, we match the user with high channel correlation to the selected cluster head user. The proposed solution is described in Algorithm 2.

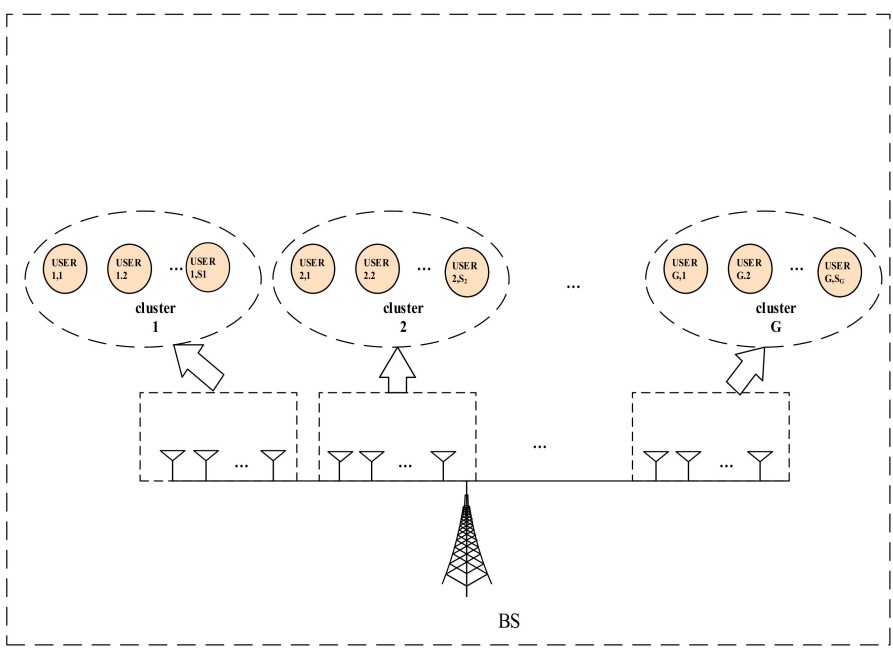

**Figure 3.** User clustering based on fixed antenna grouping.

---

**Algorithm 2**: User Clustering based on Fixed Antenna Grouping (UC_FAG)

---

Input:

The number of users $K$, and the number of antennas $N$;

Channel vectors: $\mathbf{h}_k$ for $k = 1, 2, \cdots K$ RF chains: $N_{RF}$

Output:

1. User-grounding T1

Select number of cluster-heads;

$N_t = \frac{N}{N_{RF}}$

For g=1:G

$\tilde{\mathbf{H}} = \left[\left|\tilde{\mathbf{h}}_1\right|, \left|\tilde{\mathbf{h}}_2\right|, \cdots, \left|\tilde{\mathbf{h}}_K\right|\right]$

$\tilde{\mathbf{h}}_n = \mathbf{h}_n((g-1)*N_t + 1 : g * N_t, :)$

[a,order] = (sort(H),'descend')

O(g)= [order (1)]

$\tilde{\mathbf{H}} = \tilde{\mathbf{H}}/\left|\tilde{\mathbf{h}}_{a(1)}\right|$

end

2. Include other users into each cluster;

$O^C = K/O$

$\max\left|\mathbf{h}_i^H \mathbf{h}_j\right|, \forall i \in O, j \in O^C$, Grouping the cluster channel with a high correlation

Return T1.

---

When operating the user grouping in Algorithm 1, the complexities of calculating the channel correlation and the norm channel vector are $O(KN)$ and $O(K^2N)$. The complexity of Algorithm 1 is $O(NK + K^2N)$. In Algorithm 2, the complexities of calculating channel correlation and norm effective channel vector are $O(N(K - \frac{1+N_{RF}}{2}))$ and $O\left(\frac{K^2N}{N_{RF}}\right)$. Then the complexity of Algorithm 2 is $O(N(K - \frac{1+N_{RF}}{2}) + \frac{K^2N}{N_{RF}})$. By comparing the complexity of the two algorithms, we can know that the complexity of algorithm 2 is lower than that of Algorithm 1.

*2.2. Precoding*

We consider the *n*th MIMO-NOMA cluster that contains *m* users, the channel matrix $\mathbf{H}_n \in \mathbb{C}^{m \times N}$ could be defined [24]:

$$\mathbf{H}_n = \left[\mathbf{h}_{n,1}, \mathbf{h}_{n,2}, \cdots, \mathbf{h}_{n,m}\right] \tag{7}$$

By taking the Singular Value Decomposition (SVD) of the channel matrix $\mathbf{H}_n$ we obtain:

$$\mathbf{H}_n^T = \mathbf{U}_n \sum_n \mathbf{V}_n^H \tag{8}$$

Each beam is utilized by a MIMO-NOMA cluster so that the channel corresponding to the nth beam is:

$$\tilde{\mathbf{h}}_n = \mathbf{H}_n \mathbf{u}_n^* \tag{9}$$

where $\mathbf{u}_n^*$ is the first column of $\mathbf{u}_n$. The equivalent channel matrix can be expressed as follows:

$$\tilde{\mathbf{H}} = [\tilde{\mathbf{h}}_1, \tilde{\mathbf{h}}_2, \cdots, \tilde{\mathbf{h}}_G] = [\mathbf{H}_1 \mathbf{u}_1^*, \mathbf{H}_2 \mathbf{u}_2^*, \cdots, \mathbf{H}_G \mathbf{u}_G^*] \tag{10}$$

Then, the precoding matrix can be written as:

$$\tilde{\mathbf{W}} = [\tilde{\mathbf{w}}_1, \tilde{\mathbf{w}}_2, \cdots, \tilde{\mathbf{w}}_G] = \tilde{\mathbf{H}}(\tilde{\mathbf{H}}^H \tilde{\mathbf{H}})^{-1} \tag{11}$$

After normalization of the precoding matrix, the precoding vector of the nth beam is:

$$\mathbf{w}_n = \frac{\tilde{\mathbf{w}}_n}{\left\|\tilde{\mathbf{w}}_n\right\|_2} \tag{12}$$

### 2.3. Power Allocation

In the NOMA system, the channel gain difference between users can be converted to a multiplexing gain by superposition coding. Therefore, power allocation has an important impact on system performance [25]. We proposed a dynamic power allocation method for the MIMO-NOMA system. Firstly, the transmission power is allocated according to the number of beams, which is proportional to the number of users served by the beam. Each beam is used by all users of the cluster and each MIMO-NOMA cluster contains users with near-similar channel differences. Therefore, the power allocation of users in the cluster is very important. We allocate power to users within the cluster for maximizing the cluster communication rate. The proposed power allocation method is described as:

$$P_g = P \times \frac{|S_g|}{|S_1| + |S_2| + \cdots + |S_G|} \tag{13}$$

The first step is to allocate the transmit power between beams. $P_g$ is the transmitted power in the $g$th beam, $g = 1, 2 \cdots G$. $P$ denotes the total transmitted power. After obtaining the transmit power of each beam, $S_g$ is a set of the users served by the $g$th beam. The second step is to perform power allocation on the user cluster served by the beams. We assume that the interference between users is small within the same user cluster, and the problem can be defined as:

$$\max_{p_{g,1}, p_{g,2} \cdots p_{g,S_g}} C_g = \sum_{n=1}^{|S_g|} \log_2 \left(1 + \frac{\left|\mathbf{h}_{g,n}\right|^2 p_{g,n}}{\sigma}\right)$$

$$s.t.\ C_1 : \sum_{n=1}^{S_n} p_{g,n} = P_g \tag{14}$$

where $\mathbf{h}_{g,n}$ is the channel of the $n$th user in the $g$th beam ($g = 1, 2, \cdots G, n = 1, 2, \cdots S_g$). $p_{g,n}$ denotes the transmitted power for the $n$th user in the $g$th beam. $\sigma$ denotes noise power spectral density. To solve the convex optimization problem (14), we define the Lagrange function as:

$$L(\lambda, p_{g,1}, p_{g,2} \cdots p_{g,S_g}) = \sum_{n=1}^{|S_g|} \log_2 \left(1 + \frac{\left|\mathbf{h}_{g,n}\right|^2 p_{g,n}}{\sigma}\right) + \lambda \left(\sum_{n=1}^{|S_g|} p_{g,n} - P_g\right) \tag{15}$$

where $\lambda \geq 0$,

By calculating the derivative (15):

$$\frac{\partial L}{\partial p_{g,n}} = \frac{\mathbf{h}_{g,n}}{(1 + p_{g,n}\mathbf{h}_{g,n})In2} - \lambda = 0 \tag{16}$$

we have:

$$p_{g,n} = \frac{1}{\widetilde{\lambda}} - \frac{1}{\left|\mathbf{h}_{g,n}\right|} \tag{17}$$

where $\widetilde{\lambda} = \lambda In2$, $\left|\mathbf{h}_{g,n}\right|$ is the channel gain of the $n$th user in the $g$th beam.

By substituting (17) into the constraint C1 in (14) we have:

$$\sum_{n=1}^{S_n} \frac{1}{\widetilde{\lambda}} - \frac{1}{\left|\mathbf{h}_{g,n}\right|} = P_g \tag{18}$$

$\widetilde{\lambda}$ can be written as:

$$\widetilde{\lambda} = \frac{\left|S_g\right|}{P_g + \sum_{n=1}^{\left|S_g\right|} \frac{1}{\left|\mathbf{h}_{g,n}\right|}} \tag{19}$$

Substituting (19) into (17), we have

$$p_{g,n} = \frac{P_g + \sum_{n=1}^{\left|S_g\right|} \frac{1}{\left|\mathbf{h}_{g,n}\right|}}{\left|S_g\right|} - \frac{1}{\left|\mathbf{h}_{g,n}\right|} = \frac{P_g}{\left|S_g\right|} + \frac{\sum_{n=1}^{\left|S_g\right|} \frac{1}{\left|\mathbf{h}_{g,n}\right|}}{\left|S_g\right|} - \frac{1}{\left|\mathbf{h}_{g,n}\right|} \tag{20}$$

where $P_g$ is the transmitted power in the $g$th beam, $\left|S_g\right|$ represents the number of users served by the $g$th beam, and $\left|\mathbf{h}_{g,n}\right|$ is the channel gain of the $n$th user in the $g$th beam. From (20) we obtain the transmitted power of the $n$th user in the $g$th beam and find that when the number of users in the group is larger, the power allocated to user would be reduced.

## 3. Energy Harvesting Maximizing

To maximize the harvested energy while meeting the minimum communication rate, we propose the addition of a power splitter for each user at the receiver to help implement SWIPT. This method is called SWIPT with power split [26], as shown in Figure 4. The signal received by each user is divided into two parts. One part is forwarded to the information decoder for information decoding, and the other part is subjected to Energy Harvesting (EH). The received signal to EH at the $m$th user in the $n$th beam can be can be formulated [27]:

$$y_{n,m}^{EH} = \sqrt{1 - \beta_{n,m}}\, y_{n,m} \tag{21}$$

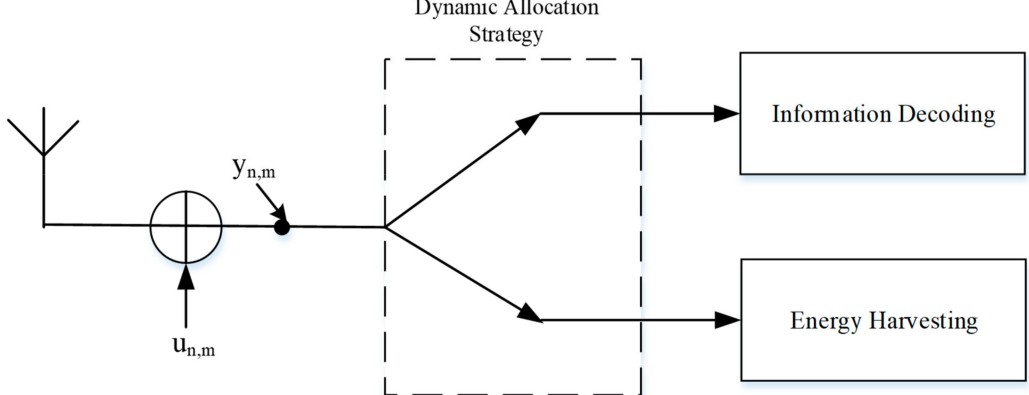

**Figure 4.** Simultaneous Wireless Information and Power Transfer (SWIPT) split receiver Power Split (PS) mode.

The harvested energy at the $m$th user in the $n$th beam is

$$P_{n,m}^{EH} = \eta(1 - \beta_{n,m})\left(\left\|\mathbf{h}_{n,m}^{H}\mathbf{w}_{\mathbf{n}}\right\|_2^2 p_{n,m} + \sigma_v^2\right) \tag{22}$$

where $\eta$ is the energy conversion efficiency, $\beta_{n,m}$ is the power splitting factor at the $m$th user in the $n$th beam, $0 \le \beta_{n,m} \le 1$.

Meanwhile, the signal used to carry out the information decoding is expressed as

$$y_{n,m}^{ID} = \sqrt{\beta_{n,m}} y_{n,m} + u_{n,m} \tag{23}$$

Substituting (1) into (23), we have

$$\begin{aligned}
y_{n,m}^{ID} = \sqrt{\beta_{n,m}} & \left( \mathbf{h}_{n,m}^H \mathbf{w}_n \sqrt{p_{m,n}} s_{m,n} + \mathbf{h}_{n,m}^H \mathbf{w}_n \sum_{j=1}^{m-1} \sqrt{p_{i,n}} s_{i,n} \right. \\
& \left. + \mathbf{h}_{n,m}^H \mathbf{w}_n \sum_{i=m+1}^{|S_i|} \sqrt{p_{i,n}} s_{i,n} + \mathbf{h}_{n,m}^H \sum_{i \ne n}^{G} \sum_{j=1}^{|S_i|} \mathbf{w}_j \sqrt{p_{i,j}} s_{i,j} + v_{n,m} \right) + u_{n,m}
\end{aligned} \tag{24}$$

where $u_{n,m}$ is the noise the distribution $CN(0, \sigma_u{}^2)$. By applying NOMA in each beam, intra-beam superposition coding of the transmitter and the receiver is realized. The $m$th user in the $n$th beam can eliminate the interference of the $i$th user (for all $i > m$) in the $n$th beam by performing SIC, and the remaining received signal of the $m$th user in the $n$th beam to information decoding can be rewritten as

$$\hat{y}_{n,m}^{ID} = \sqrt{\beta_{n,m}} \left( \mathbf{h}_{n,m}^H \mathbf{w}_n \sqrt{p_{n,m}} s_{n,m} + \mathbf{h}_{n,m}^H \mathbf{w}_n \sum_{j=1}^{m-1} \sqrt{p_{i,n}} s_{i,n} + \mathbf{h}_{n,m}^H \sum_{i \ne n}^{G} \sum_{j=1}^{|S_i|} \mathbf{w}_j \sqrt{p_{i,j}} s_{i,j} + v_{n,m} \right) + u_{n,m} \tag{25}$$

Then, according to (25), the SINR at the $m$th user in the $n$th beam can be written as

$$\gamma_{n,m} = \frac{\left\| \mathbf{h}_{n,m}^H \mathbf{w}_n \right\|_2^2 p_{n,m}}{\xi_{n,m}} \tag{26}$$

where,

$$\xi_{n,m} = \left\| \mathbf{h}_{n,m}^H \mathbf{W}_n \right\|_2^2 \sum_{j=1}^{m-1} p_{n,j} + \sum_{i \ne n}^{G} \left\| \mathbf{h}_{n,m}^H \mathbf{W}_i \right\|_2^2 \sum_{j=1}^{|S_i|} p_{i,j} + \sigma_v^2 + \frac{\sigma_u^2}{\beta_{n,m}} \tag{27}$$

The achievable rate of the $m$th user in the $n$th beam can be written as

$$R_{n,m} = \log_2(1 + \gamma_{n,m}) \tag{28}$$

We have grouped users, designed a precoding matrix, and allocated power to users in Sections 2 and 3. According to (22), we know we need to find the power splitting coefficient of each user for making the harvested energy at the receiver is maximized. We formulate the problem as

$$\begin{aligned}
& \max_{\{\beta_{n,m}\}} P^{EH} \\
& s.t. \; C_1 : R_{m,n} \ge R_{\min} \\
& \qquad C_2 : 0 \le \beta_{n,m} \le 1
\end{aligned} \tag{29}$$

Substituting (22), (28), into (29), we have

$$\begin{aligned}
& \max_{\{\beta_{n,m}\}} \sum_{i=1}^{G} \sum_{j=1}^{|S_j|} \eta(1 - \beta_{n,m}) \left( \left\| \mathbf{h}_{n,m}^H \mathbf{w}_n \right\|_2^2 p_{n,m} + \sigma_v{}^2 \right) \\
& s.t. \; C_1 : \log_2(1 + \gamma_{n,m}) \ge R_{\min} \\
& \qquad C_2 : 0 \le \beta_{n,m} \le 1
\end{aligned} \tag{30}$$

where $P^{EH}$ is the total harvested energy. $R_{\min}$ denotes the minimum achievable rate of the user.

To maximize the total harvested energy, the energy harvested by each user is maximal, the problem is converted to maximize the energy harvested by each user:

$$
\begin{aligned}
&\max_{\{\beta_{n,m}\}} P_{n,m} = \eta(1-\beta_{n,m})\left(\left\|\mathbf{h}_{n,m}^{H}\mathbf{w}_n\right\|_2^2 p_{n,m} + {\sigma_v}^2\right) \\
&s.t.\ \mathrm{C}_1: \log_2(1+\gamma_{n,m}) \geq R_{\min} \\
&\qquad \mathrm{C}_2: 0 \leq \beta_{n,m} \leq 1
\end{aligned}
\tag{31}
$$

Substituting (26), (27) into (31), we have

$$
\begin{aligned}
&\max_{\{\beta_{n,m}\}} P_{n,m} = \eta(1-\beta_{n,m})\left(\left\|\mathbf{h}_{n,m}^{H}\mathbf{w}_n\right\|_2^2 p_{n,m} + {\sigma_v}^2\right) \\
&s.t.\ \mathrm{C}_1: \log_2\left(1 + \frac{\left\|\mathbf{h}_{n,m}^{H}\mathbf{w}_n\right\|_2^2 p_{n,m}}{\left\|\mathbf{h}_{n,m}^{H}\mathbf{W}_n\right\|_2^2 \sum\limits_{j=1}^{m-1} p_{n,j} + \sum\limits_{i\neq n}^{G}\left\|\mathbf{h}_{n,m}^{H}\mathbf{W}_i\right\|_2^2 \sum\limits_{j=1}^{|S_i|} p_{i,j} + \sigma_v^2 + \frac{\sigma_u^2}{\beta_{n,m}}}\right) \geq R_{\min} \\
&\qquad \mathrm{C}_2: 0 \leq \beta_{n,m} \leq 1
\end{aligned}
\tag{32}
$$

By simplifying C1 in (32),

$$
1 + \frac{\left\|\mathbf{h}_{n,m}^{H}\mathbf{w}_n\right\|_2^2 p_{n,m}}{\omega_{n,m} + \frac{\sigma_u^2}{\beta_{n,m}}} \geq 2^{R_{\min}}
\tag{33}
$$

where,

$$
\omega_{n,m} = \left\|\mathbf{h}_{n,m}^{H}\mathbf{W}_n\right\|_2^2 \sum_{j=1}^{m-1} p_{n,j} + \sum_{i\neq n}\left\|\mathbf{h}_{n,m}^{H}\mathbf{W}_i\right\|_2^2 \sum_{j=1}^{|S_i|} p_{i,j} + \sigma_v^2
\tag{34}
$$

By simplifying (33),

$$
\beta_{n,m} \geq \frac{\sigma_u^2(2^{R_{\min}}-1)}{\left\|\mathbf{h}_{n,m}^{H}\mathbf{w}_n\right\|_2^2 p_{n,m} - \omega_{n,m}(2^{R_{\min}}-1)}
\tag{35}
$$

Then, the constraint $\mathrm{C}_1$ in (32) can be rewritten as

$$
\begin{aligned}
&\max_{\{\beta_{n,m}\}} P_{n,m} = \eta(1-\beta_{n,m})\left(\left\|\mathbf{h}_{n,m}^{H}\mathbf{w}_n\right\|_2^2 p_{n,m} + {\sigma_v}^2\right) \\
&s.t.\ \mathrm{C}_1: \beta_{n,m} \geq \frac{\sigma_u^2(2^{R_{\min}}-1)}{\left\|\mathbf{h}_{n,m}^{H}\mathbf{w}_n\right\|_2^2 p_{n,m} - \omega_{n,m}(2^{R_{\min}}-1)} \\
&\qquad \mathrm{C}_2: 0 \leq \beta_{n,m} \leq 1
\end{aligned}
\tag{36}
$$

According to (36), we know that when $\beta_{n,m}$ is the minimum that meets constraint $\mathrm{C}_1$, $1-\beta_{n,m}$ is the maximum. Then we obtain the maximum $P_{n,m}$. Accordingly, we get the optimal power splitting coefficient at the *m*th user in the *n*th beam:

$$
\beta_{n,m} = \frac{\sigma_u^2(2^{R_{\min}}-1)}{\left\|\mathbf{h}_{n,m}^{H}\mathbf{w}_n\right\|_2^2 p_{n,m} - \omega_{n,m}(2^{R_{\min}}-1)}
\tag{37}
$$

Substituting (37) into (22), The maximal harvested energy at the *m*th user in the *n*th beam is

$$
P_{n,m_{\max}} = \eta\left(1 - \frac{\sigma_u^2(2^{R_{\min}}-1)}{\left\|\mathbf{h}_{n,m}^{H}\mathbf{w}_n\right\|_2^2 p_{n,m} - \omega_{n,m}(2^{R_{\min}}-1)}\right)\left(\left\|\mathbf{h}_{n,m}^{H}\mathbf{w}_n\right\|_2^2 p_{n,m} + {\sigma_v}^2\right)
\tag{38}
$$

## 4. Simulation Results

We consider a typical downlink mmWave massive MIMO-NOMA system, the spectral efficiency is defined as the reachability rate in equation (6), and the energy efficiency is defined as the ratio of reachability to total power consumption [28],

$$EE = \frac{R_{sum}}{P + N_{RF}P_{RF} + P_{BB}} \tag{39}$$

where $P$ is the total transmitted power,$P_{RF}$ is the power consumed by each RF chain, $P_{BB}$ is the baseband power consumption, $N_{RF}$ is the number of the RF chain, $R_{sum}$ is from (6).Simulation parameters are shown in Table 1.

**Table 1.** Simulation parameters.

| Parameter | Value |
|---|---|
| Number of antennas at BS | 256 |
| Number of RF chain | 8 |
| Users | 32 |
| Antenna number of each user | 1 |
| Total transmitted power P (mW) | 32 |
| Minimum user communication rate (bps/Hz) | 0.3 |
| Each RF chain power consumption, P (mW) | 300 |
| Baseband power consumption, P (mW) | 200 |

In the simulation, we first consider three kinds of mmWave massive MIMO systems and compare them by using two different user grouping methods proposed in Section 2.1:

(1) "Full-digital MIMO system" with one RF chain connected to each antenna ($N=N_{RF}$).
(2) "MIMO-NOMA under the UC_CG algorithm" grouping users according to the proposed UC_CG algorithm, and performing NOMA for the user in the beam.
(3) "MIMO-NOMA under the UC_FAG algorithm" grouping users according to the proposed UC_FAG algorithm and performs NOMA for the user in the beam.
(4) MIMO-OMA under the UC_CG algorithm": The user is grouped according to the UC_CG algorithm, and the OMA is executed for the user in the beam.
(5) "MIMO-OMA under the UC_FAG algorithm": The users are grouped according to the proposed UC_FAG algorithm, and the OMA is executed for the user in the beam.

The power allocation method proposed in this paper is applied to the MIMO-NOMA system and we compare the performance with the system using traditional average power allocation method that it allocated equal power to all users. Finally, the SWIPT technology is integrated into the system to compare the power harvested by the MIMO-NOMA and MIMO-OMA.

Figure 5 shows the spectral efficiency against Signal to Noise Ratio(SNR)of the considered five schemes mentioned above, where the number of users *K* is set to 32 and the number of antennas is set to 256. From the figure, we can see that the proposed MIMO-NOMA scheme has a higher spectral efficiency than the MIMO-OMA scheme. It is intuitive that the fully digital MIMO can achieve the best spectrum efficiency, as shown in Figure 5. However, the number of RF chains required in the full digital MIMO scheme is equal to the number of antennas ($N_{RF} = N$), and the number of RF chains required in MIMO-NOMA is 8. The full digital MIMO scheme needs higher hardware costs and overhead. Through the simulation diagram, we can obtain that the UC_CG algorithm gives higher spectral efficiency than the UC_FAG algorithm. Given the users that are matched according to the correlation between all channels of the user and the cluster-head user in UC_CG algorithm, in the UC_FAG algorithm, the users are matched according to the correlation between the part of the channel

of the user and the cluster-head user. In comparison with the UC_FAG algorithm, the interference between users in the group is smaller in the UC_CG algorithm.

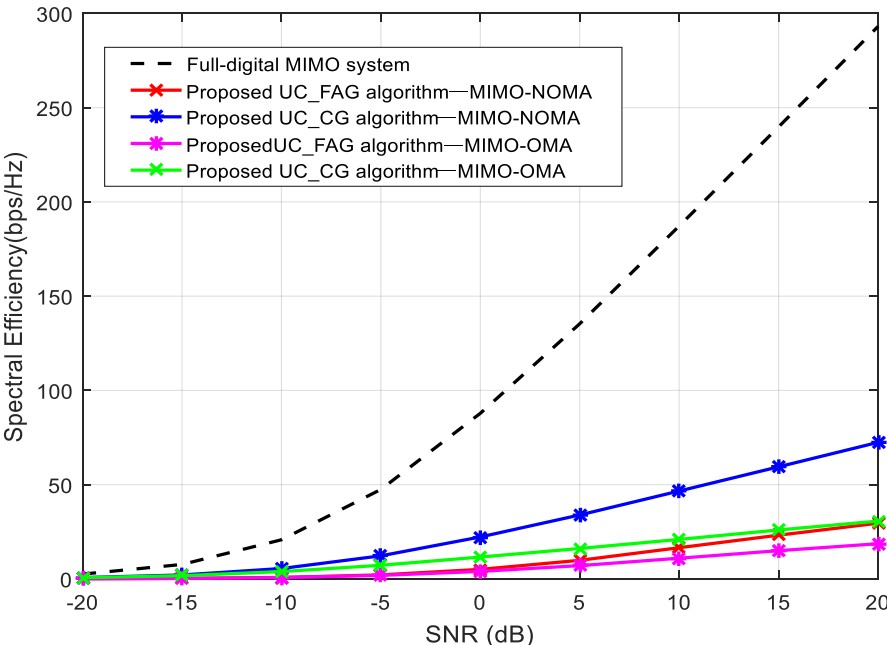

**Figure 5.** Spectral efficiency of the considered five schemes system against SNR

Figure 6 shows the energy efficiency of the five schemes considered under different SNR, where the number of users is set to 32, the number of antennas is set to 256. From Figure 6, we know that the MIMO-NOMA scheme has a higher energy efficiency than MIMO-OMA and fully digital MIMO, where the number of RF chains of the fully digital MIMO is equal to the number of base station antennas, which results in very high energy consumption. In contrast, in the MIMO-NOMA scheme, the number of RF chains is much smaller than the number of antennas. Therefore, the energy consumption of the RF chain can be significantly reduced when compared with the fully digital MIMO scheme.

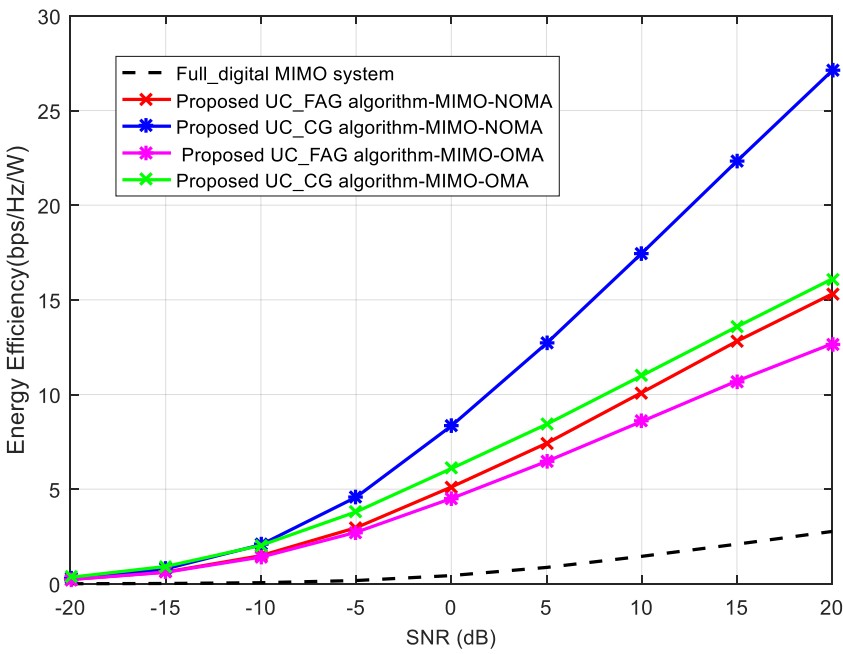

**Figure 6.** Energy efficiency against SNR

According to Figure 5, the communication rates of UC_CG with MIMO-NOMA is higher than UC_FAG with MIMO-NOMA. When the total power consumption of the system is the same, UC_CG with MIMO-NOMA has higher energy efficiency than UC_FAG with MIMO-NOMA according to (38). Therefore, when compared with other four schemes, the energy efficiency of UC_CG with MIMO-NOMA is the highest.

A comparison of the performance of energy efficiency with the number of users is shown in Figure 7 in which the SNR is set to 10dB. We can see that, as the number of users increases, the energy efficiency is gradually reduced. Even with a very large number of users, the proposed MIMO-NOMA scheme is more energy efficient than MIMO-OMA and the fully digital MIMO scheme.

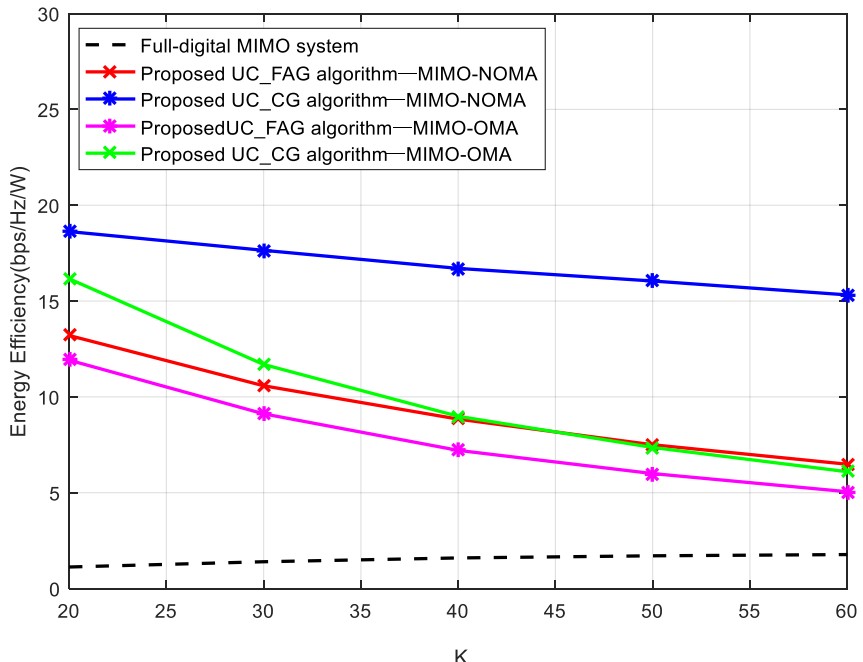

**Figure 7.** Energy efficiency against the number of users K.

The next experiment considers the spectral efficiency of the SNR under two different power allocation algorithms. From Figure 8, we obtain that the power allocation algorithm proposed in this paper has higher spectrally efficient than the traditional average power allocation algorithm. We understand that the power allocation algorithm proposed is better than the traditional average allocation algorithm.

Figure 9 shows the energy harvesting performance against SNR. To enable the user to maximize harvested power and meet the communication requirement, in Section 3, we proposed a method that finds the power splitting optimization. From Figure 9, we can see that, when signal power is low, the received signal performs information decoding. The receiver can start harvesting energy when the signal becomes larger. In comparison with the MIMO-OMA scheme, MIMO-NOMA can harvest more energy. Therefore, the proposed MIMO-NOMA scheme with SWPIT is superior to MIMO-OMA scheme, which can realize the recycling of energy.

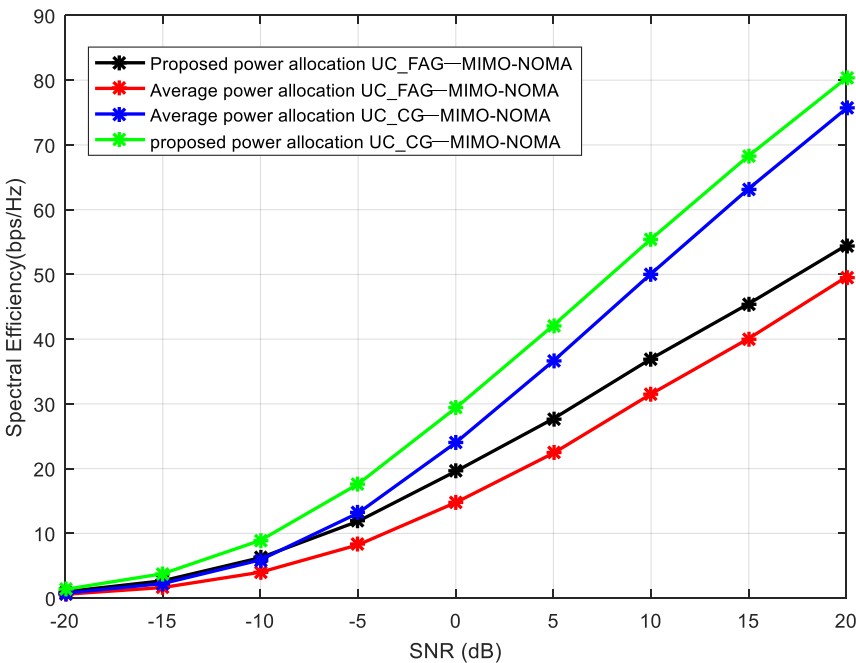

**Figure 8.** Energy efficiency of the different power allocation methods against SNR.

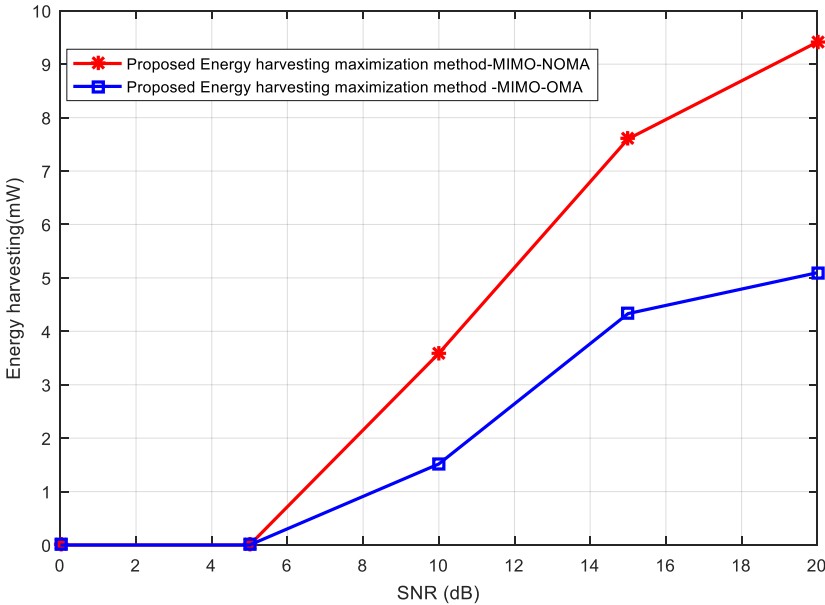

**Figure 9.** Energy harvesting against SNR.

## 5. Conclusions

In this paper, we designed two different user grouping methods for MIMO-NOMA system: the UC_CG algorithm and the UC_FAG algorithm. From the simulation, we can see that the UC_CG algorithm is better than the UC_FAG algorithm, which improves spectral efficiency. We proposed a new power allocation method. The simulation results show that the algorithm is superior to the traditional average power allocation algorithm. Finally, we apply the SWIPT for MIMO-NOMA system. Under the premise of satisfying the minimum communication rate of each user, we proposed the method based on maximizing the harvested energy to find the optimal power splitting factor for each user. This method allows the system to harvest more energy and meets the user's minimum communication rate, thereby achieving the recycling of energy and green communication.

**Author Contributions:** Conceptualization, S.L.; Methodology, Z.W.; Supervision, L.J. and J.D.; Writing—review & editing, S.L. All authors have read and agreed to the published version of the manuscript.

**Funding:** This work was supported by National Nature Science Funding of China (NSFC): 61401407, 61601414 and the Fundamental Research Funds for the Central Universities.

**Conflicts of Interest:** The authors declare no conflict of interest.

## Abbreviations

The following abbreviations are used in this manuscript:

| | |
|---|---|
| MIMO | Multiple Input Multiple Output |
| NOMA | Non-Orthogonal Multiple Access |
| SIC | Serial Interference Cancellation |
| SWIPT | Simultaneous Wireless Information and Power Transfer |
| OMA | Orthogonal Multiple Access |
| RF | Radio Frequency |
| EH | Energy Harvesting |
| UC_CG | User Clustering based on Channel Gain UC_CG |
| UC_FAG | User Clustering based on Fixed Antenna Grouping |
| SINR | Signal to Interference plus Noise Ratio |
| SNR | Signal to Noise Ratio |
| PS | Power Split |
| BS | Base Station |

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
