# Peer review of "Energy Harvesting Maximizing for Millimeter-Wave Massive MIMO-NOMA"

_electronics, doi:10.3390/electronics9010032_

Round 1

Reviewer 1 Report

Your paper proposes novel ideas for MIMO-NOMA, which is an important topic along with cognitive radio for 5G mmWave. Energy harvesting from 5G mmWave seems too ambitious from circuit design point of view. Please improve figures and grammar. 

Author Response

The authors would like to thank all the reviewers for their constructive comments and suggestions. We have thoroughly revised the paper to address all the concerns raised by reviewers. Below, we provide a detailed response to comments proposed by reviewer 1 and the comments are written in Italic font. We hope you would satisfy with the revision.

Comment 1. Your paper proposes novel ideas for MIMO-NOMA, which is an important topic along with cognitive radio for 5G mmWave. Energy harvesting from 5G mmWave seems too ambitious from circuit design point of view. Please improve figures and grammar.

Response: Thank you for your positive evaluation along with comments and suggestions to improve our manuscript. We have carefully checked this paper and carefully revised the grammar and spelling errors. We examined and revised many sentences to make the paper more readable. We have checked all figures and modified errors in figures to improve the quality of the figures. We have added formula derivation process and added definition of notations and abbreviations.

Reviewer 2 Report

Although overall contents is written well, it should be revised in terms of clear logic and explanation. 

From Eq. (20), it needs to add description about relation among allocated power, set of users, channel.

In Eq.(30), 'substituting (21),(27)' should be fixed as (22),(28).

It needs to add additional information how (32) is obtained by simplifying (31).

Could you explain meaning of Eq.(35)?

In Figure 2, cluster 2 should be revised as cluster 3.

In this paper, there are so many English expression and grammar errors. For example, the subject and verb don't match in many sentences. In addition, there are many typing errors. (ex. procoding, percoding) Authors have to fix all errors correctly. 

What does 'number of each user : 1' mean?

Why is total Tx power is much smaller than power consumed by RF chain and baseband power consumption? It needs more explanations in simulation parameters.

What is traditional average power allocation algorithm?

In Fig. 5, it's necessary to add logical reason why UC_CG algorithm has a better spectral efficiency than UC_FAG algorithm. The complexity comparison is also required.

In Fig 6, logical description needs why energy efficiency of UC_CG with MIMO-NOMA is highest.

Author Response

The authors would like to thank all the reviewers for their constructive comments and suggestions. We have thoroughly revised the paper to address all the concerns raised by reviewers. Below, we provide a detailed response to comments proposed by reviewer 2 and the comments are written in Italic font. We hope you would satisfy with the revision.

Major Comments

Although overall contents is written well, it should be revised in terms of clear logic and explanation.

Response: Thank you for your positive evaluation along with comments and suggestions to improve our manuscript. Our revision and responses to the concerns you raised are provided below. Besides, we thoroughly examined and revised the manuscript.

Comment 1. From Eq. (20), it needs to add description about relation among allocated power, set of users, channel.

Response: Thank you for pointing it out. We have added the description under the (20) and explained the notations.

Comment 2. In Eq.(30), 'substituting (21),(27)' should be fixed as (22),(28).

Response: Thank you for pointing out. We have modified the sentence: 'substituting (21),(27)' to 'substituting (22),(28)' and checked other expressions.

Comment 3. It needs to add additional information how (32) is obtained by simplifying (31).

Response: Thank you for raising this issue. We have added detailed steps to show how (32) is obtained by simplifying (31). Detailed steps are presented in (32)-(36).

Comment 4. Could you explain meaning of Eq.(35)?

Response: Thank you for raising this issue. We have added the explanation above the Eq(37).

Comment 5. In Figure 2, cluster 2 should be revised as cluster 3.

Response: We are sorry for our careless mistakes. Thank you for your reminding. We have modified Figure 2 and changed cluster 2 to cluster 3. We also checked other figures to make the same problem doesn't occur.

Comment 6. In this paper, there are so many English expression and grammar errors. For example, the subject and verb don't match in many sentences. In addition, there are many typing errors. (ex. procoding, percoding) Authors have to fix all errors correctly.

Response: Thank you for raising this issue. We have carefully checked this paper and carefully revised the grammar and spelling errors.

Comment 7. What does 'number of each user : 1' mean?

Response: Thank you for pointing out. We want to express the antenna number of each user is 1, so we have revised the expression: 'number of each user : 1' to ' antenna  number of each user : 1'.

Comment 8. Why is total Tx power is much smaller than power consumed by RF chain and baseband power consumption? It needs more explanations in simulation parameters.

Response: In order to meet the high data throughput for users in massive MIMO, high-power RF chains and basebands are generally selected .According to references ‘L. Dai, B. Wang, M. Peng and S. Chen. Hybrid Precoding-Based Millimeter-Wave Massive MIMO-NOMA With Simultaneous Wireless Information and Power Transfer.  IEEE Journal on Selected Areas in Communications .Jan. 2019,vol. 37, no. 1, pp. 131-141.’, we adopt the typical values PRF = 300 mW and PBB = 200 mW. We also have added more explanations in simulation parameters.

Comment 9. What is traditional average power allocation algorithm?

Response: We have explained the traditional average power allocation algorithm in paper.

Comment 10. In Fig. 5, it's necessary to add logical reason why UC_CG algorithm has a better spectral efficiency than UC_FAG algorithm. The complexity comparison is also required.

Response: Thank you for the suggestion. We have added the add logical reason in 4 Simulation Results. Due to the users are matched according to the correlation between all channels of the user and the cluster-head user in UC_CG algorithm, while in UC_FAG algorithm the users are matched according to the correlation between the part of the channel of the user and the cluster-head user. Compared with UC_FAG algorithm, the interference between users in the group in UC_CG algorithm is smaller. We have added the complexity comparison in Section 2.1.

Comment 10. In Fig 6, logical description needs why energy efficiency of UC_CG with MIMO-NOMA is highest.

Response: Thank you for pointing out. We have added the logical description in paper, according to Figure 5 the communication rates of UC_CG with MIMO-NOMA is higher than UC_FAG with MIMO-NOMA. When the total power consumption of the system is the same, U C_CG with MIMO-NOMA has higher energy efficiency than UC_FAG with MIMO-NOMA according to (39). Therefore UC_CG with MIMO-NOMA compared to other four schemes, the energy efficiency is highest.

Round 2

Reviewer 1 Report

It must be noted that energy harvesting from millimeter waves is much harder than sub-6GHz regime and has low efficiency, however, if strong beamforming is used in 5G mmWave to focus power, the idea of MIMO-NOMA can be useful for cognitive radio with shared spectrum.

Author Response

The authors would like to thank all the reviewers for their constructive comments and suggestions. We have thoroughly revised the paper to address all the concerns raised by reviewers. Below, we provide a detailed response to comments proposed by reviewer 1 and the comments are written in Italic font. We hope you would satisfy with the revision.

Comment 1. It must be noted that energy harvesting from millimeter waves is much harder than sub-6GHz regime and has low efficiency, however, if strong beamforming is used in 5G mmWave to focus power, the idea of MIMO-NOMA can be useful for cognitive radio with shared spectrum.

Response: Thank you for pointing out. We strongly agree with your view. As your point of view that if strong beamforming is used in 5G mmWave to focus power, the MIMO-NOMA technology can be used very well in cognitive radio with shared spectrum. So, our future work will study how to use strong beamforming in 5G mmWave to focus power.

Reviewer 2 Report

Most of revisions have been well performed. However, it still needs to fix some expression errors.

In Fig.2, cluster 2 should be corrected as cluster 3.

In line 287, expression should be revised as 'By simplifying C1 in (32)'.

In line 291, expression should be revised as 'By simplifying (33)'.

In line 323 and 328, MIMO-MOMA should be corrected as MIMO-NOMA.

In line 188, 'Thun' should be corrected as 'Then'. There is no sentence which complexity of Alg. 2 is lower than that of Alg. 1.

Author Response

The authors would like to thank all the reviewers for their constructive comments and suggestions. We have thoroughly revised the paper to address all the concerns raised by reviewers. Below, we provide a detailed response to comments proposed by reviewer 2 and the comments are written in Italic font. We hope you would satisfy with the revision.

Comment 1. In Fig.2, cluster 2 should be corrected as cluster 3.

Response: We are sorry for our careless mistakes. Thank you for your reminding. We have modified Figure 2 and changed cluster 2 to cluster 3.

Comment 2. In line 287, expression should be revised as 'By simplifying C1 in (32)'.

Response: Thank you for pointing out. We have modified the expression: ‘Simplify in (32)have:’ to 'By simplifying C1 in (32)’.

Comment 3. In line 291, expression should be revised as 'By simplifying (33)'.

Response: Thank you for pointing out. We have modified the expression to 'By simplifying (33)'.

Comment 4. In line 323 and 328, MIMO-MOMA should be corrected as MIMO-NOMA.

Response: We are sorry for our careless mistakes. Thank you for your reminding. We have modified ‘MIMO-MOMA’ to ‘MIMO-NOMA’

Comment 5. In line 188, 'Thun' should be corrected as 'Then'. There is no sentence which complexity of Alg. 2 is lower than that of Alg. 1

Response: Thank you for pointing out. We have modified ‘Thun’ to ‘Then’ and added the sentence: ‘By comparing the complexity of the two algorithms, we can know that the complexity of algorithm 2 is lower than that of algorithm 1’ in line 189.
